# Compliant Residual DAgger: Improving Real-World Contact-Rich Manipulation with Human Corrections

Xiaomeng Xu[*]    Yifan Hou[*]    Zeyi Liu    Shuran Song

Stanford University

## Abstract

We address key challenges in Dataset Aggregation (DAgger) for real-world contact-rich manipulation: how to collect informative human correction data and how to effectively update policies with this new data. We introduce Compliant Residual DAgger (CR-DAgger), which contains two novel components: 1) a Compliant Intervention Interface that leverages compliance control, allowing humans to provide gentle, accurate delta action corrections without interrupting the ongoing robot policy execution; and 2) a Compliant Residual Policy formulation that learns from human corrections while incorporating force feedback and force control. Our system significantly enhances performance on precise contact-rich manipulation tasks using minimal correction data, improving base policy success rates by over 60% on two challenging tasks (book flipping and belt assembly) while outperforming both retraining-from-scratch and finetuning approaches. Through extensive real-world experiments, we provide practical guidance for implementing effective DAgger in real-world robot learning tasks. Result videos are available at: https://compliant-residual-dagger.github.io/

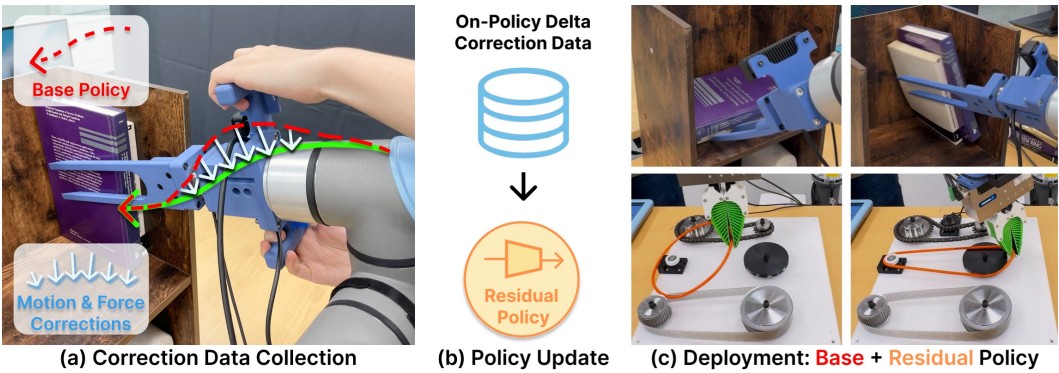

(a) Correction Data Collection    (b) Policy Update    (c) Deployment: Base + Residual Policy

Figure 1: **CR-DAgger.** To improve a robot manipulation policy, we propose a compliant intervention interface (a) for collecting human correction data, and use this data to update a compliant residual policy (b), and thoroughly study their effects by deploying the updated policy on two contact-rich manipulation tasks in the real world (c).

## 1 Introduction

Learning from human demonstrations has seen many recent successes in real-world robotic tasks [9, 21, 41, 48, 42]. However, to obtain a successful policy, human demonstrators often have to repeatedly deploy a policy and observe its failure cases, then collect more data to update the policy until it

---

[*]Equal Contributions

39th Conference on Neural Information Processing Systems (NeurIPS 2025).

succeeds. This process is broadly referred to as Dataset Aggregation (DAgger) [36, 23]. However, doing DAgger effectively for real-world robotic problems still faces the following challenges:

**How to collect informative human correction data?** DAgger is most effective when the correction data is within the original policy's induced state-action distribution [36]. In practice, the common approach is either (1) collecting offline demonstrations that cover the policy's typical failure scenarios [8], or (2) human taking over robot control during policy deployment [37, 32]. However, in both cases, the human demonstrator has no access to the original policy's behavior and may deviate excessively from it. Human taking over additionally introduces force discontinuity when they do not instantly reproduce the exact same robot force. This is partially due to the lack of effective correction interfaces that support precise and instantaneous intervention.

**How to effectively update the policy with new data?** Prior methods for improving a pretrained policy with additional data include (1) retraining the policy from scratch with the aggregated dataset [23], which can be computationally expensive; (2) finetuning the policy with only the additional data [41, 17, 7], which is sensitive to the quality of the new data [51], and (3) training a residual policy separately on top of the pretrained policy, which is typically done with Reinforcement Learning [2, 51] or Imitation Learning [5], both require a large number of samples.

In this work, we address these questions by proposing an improved system **Compliant Residual DAgger (CR-DAgger)** consisting of two critical components:

- **Compliant Intervention Interface.** We propose an on-policy correction system based on kinesthetic teaching to collect delta action *without interrupting the current robot policy*. Leveraging compliance control, the interface lets humans directly apply force to the robot and feel the magnitude of their instantaneous correction. Unlike take-over corrections, our design allows smooth transition between correction/no correction mode, while providing direct control of correction magnitudes.
- **Compliant Residual Policy.** Leveraging the force feedback from our Compliant Intervention Interface, we propose a residual policy formulation that takes in an *extra force modality* and predicts both residual motions and *target forces*, which can fully describe the human correction behavior. The Compliant Residual Policy is force-aware, even when the base policy is position-only. We show that our residual policy formulation *learns effective correction strategies* using the data collected from our Compliant Intervention Interface.

Together, our system significantly improves the success rate of precise contact-rich robot manipulation tasks using a small amount of additional data. We demonstrate the efficacy of our method on two challenging tasks involving long horizons and sequences of contacts: book flipping and belt assembly. We improve over the base policy success rate by over 60%, while also outperforming retrain-from-scratch and finetuning under the same data budgets. In summary, our contributions are:

- A **Compliant Intervention Interface**, a system that allows humans to provide accurate, gentle, and smooth corrections in both position and force to a running robot policy without interrupting it.
- A **Compliant Residual Policy**, a policy formulation that seamlessly integrates additional force modality inputs and predicts residual motions and forces.
- A **practical guide** for efficient DAgger based on extensive real-world studies for critical but often overlooked design choices, such as batch size and sampling strategy. Our hardware design, training data, and policy code can be found here.

## 2 Related Work

**Human-in-the-Loop Corrections for Robot Policy Learning**. The original DAgger work [36] requires the demonstrator to directly label actions generated by the policy. In robotics, a practical variation is to let the human *take over* the robot control and provide correct action directly [23]. Such human correction motion can be recorded with spacemouse [7, 29], joystick [37], smartphone [32], or arm-based teleoperation system [18, 19, 41]. We instead proposes a novel kinesthetic teaching system with compliance controller that allows the demonstrator to apply delta corrections while the robot policy is still running, and additionally records force feedback. Our results show that both the delta correction data and the force data are crucial to the success of the learned policy.

**Improving Pretrained Robot Policies with New Data**. The most direct approach to improving a pretrained policy with new correction data is to retrain the policy on the aggregated dataset, combining

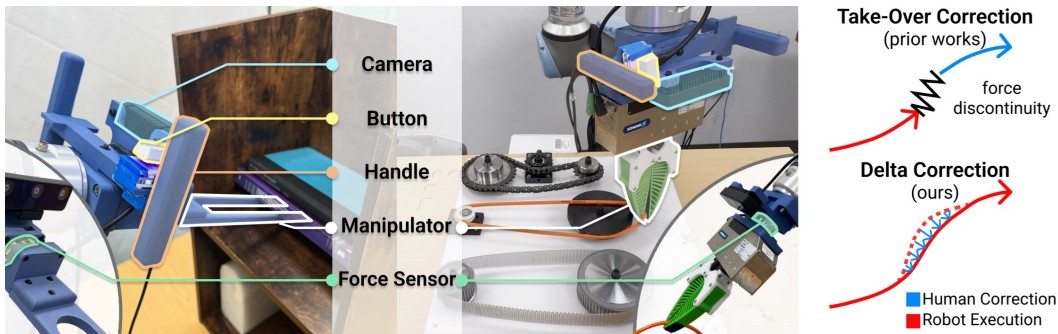

Figure 2: **Compliant Intervention Interface** characterized by a kinesthetic correction hardware setup where humans hold on the handle and apply forces to correct robot execution, providing on-policy delta corrections.

prior demonstrations with new feedback [32, 37]. Alternatively, Reinforcement Learning (RL) offers a framework to incorporate both offline and online data, either by warm-starting replay buffers [31, 3] or by using offline data to guide online fine-tuning [49, 43]. When policies are trained on large-scale human demonstration datasets [6, 38, 25, 34, 24, 44, 45], retraining becomes impractical, especially when the original data is inaccessible. In such cases, fine-tuning with only the new data is a common solution, using either imitation learning [29, 17, 41] or RL [33, 7]. Another line of work introduces an additional residual model on top of the original policy. These residual policies can be trained with RL in simulation [51, 2, 15], but suffers from sim-to-real challenges. Training residual policy in the real world usually requires a large number of samples [5, 22], intermediate scene representation [14], or consistent visual observations between training and testing [13, 16], making the approach hard to adopt in practice. In this work, we introduce a practical data collection system and an efficient residual policy learning algorithm for long-horizon, contact-rich manipulation tasks. Our approach requires only a small amount of real-world correction data and supports integration of additional sensory modalities not present in the original model, leading to improved policy performance.

## 3  CR-DAgger Method

Our goal is to improve a pretrained robot policy with a small amount of human correction data. To achieve this, we propose a Compliant Intervention Interface (§ 3.1) that enables precise and intuitive on-policy human correction data collection, and a Compliant Residual Policy (§ 3.2) that efficiently learns the correction behaviors to be used on top of the pretrained policy. Throughout the paper, we use the term *base policy* to refer to the pretrained policy without online improvements.

### 3.1  Compliant Intervention Interface

Correction data is collected by human demonstrators to rectify policy failures. Unlike initial demonstrations that establish baseline behaviors, correction data specifically targets failure modes observed during policy deployment. Correction data is most effective when it corrects failures in policy-induced state distributions [36]. The interface through which these corrections are collected significantly impacts the quality of correction data, which should be intuitive for demonstrators, capture critical corrective information at precise moments of failures, and facilitates collecting data close to the base policy state-action distribution.

There are two types of correction collection methods: *Off-policy correction* is when humans observe failures of the base policy during deployment, and then recollect extra offline demonstrations to address failure cases. This approach is most commonly used for improving Behavior Cloning policy performance due to its simplicity - it requires *no additional infrastructure* beyond the original data collection setup. However, the resulting demonstrations may fail to cover all the failure cases or deviate from the original state-action distribution. We focus on *on-policy correction* instead, where humans can monitor policy execution and intervene on the spot when failures occur or are anticipated. This approach allows humans to provide corrections more targeted to the base policy's failure cases. However, challenges still exist for an intervention system:

- **Non-smooth transitions**. Intervention in robotics is typically implemented by *take-over* correction: letting human take complete control and overwrite robot policy. As the underlying control abruptly switches between robot policy and human intention, disturbances are introduced due to policy

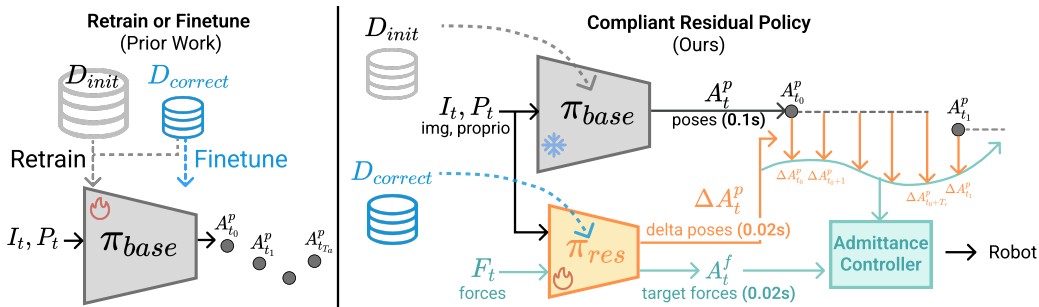

Figure 3: **Policy Update Methods.** Left: Common policy update methods - retraining and finetuning. Right: Ours. The base policy runs at 1 Hz. It takes in images $I_t$ and proprioceptions $P_t$ and predicts 32 frames of end-effector poses $A_t^p = \{A_{t_0}^p, A_{t_1}^p, ..., A_{t_{T_a}}^p\}$ spaced 0.1 Seconds apart. The Compliant Residual Policy runs at 50 Hz. It takes in additional force inputs $F_t$ and predicts 5 frames of delta poses $\Delta A_t^p = \{\Delta A_{t_0}^p, \Delta A_{t_0+1}^p, ..., \Delta A_{t_0+T_r}^p\}$ and target forces $A_t^f$ spaced 0.02 Seconds apart. The combined poses of $A_t^p$ and $\Delta A_t^p$, and target forces $A_t^f$ are taken by an admittance controller to command the robot.

inference and human response latency, especially when the robot is withholding external forces. The recorded data thus may include undesired actions that do not reflect the human's intention.

- **Distribution shift.** The human-intervened state-action distribution may deviate significantly from the original distribution. Additionally, the non-smooth transition above could bring in disturbances and add to the distribution shift.

- **Indirect correction brings errors.** Correction is commonly implemented via teleoperation interfaces such as spacemouse or joysticks [17, 7]. With spatial mismatch and teleoperation latency, it is hard for the demonstrator to instantly provide accurate corrections upon intervention starts without going through a short adjustment period.

- **Missing information.** The recorded correction data need to fully describe the human's intended action. Simply recording the robot's position is not sufficient, since it may be under the influence of human correction force and will cause different result when testing without human.

We propose a *Compliant Intervention Interface* with the following designs to solve those challenges:

- **Delta correction instead of take-over correction.** Unlike take-over correction, where the demonstrator has no idea of the policy's original intention once taking over, we propose a novel on-policy delta correction method: we let the robot policy executes continuously while the human applies forces to the robot with a handle mounted on the end effector, resulting in delta actions on top of the policy action. The human demonstrator can always sense the policy's intention through haptic feedback, and easily control the magnitude of intervention by the amount of force applied to the handle. As a result, delta correction ensures smooth intervention data and limits the human from providing very large corrections. The approach is also intuitive as the human can directly move the robot towards desired correction directions.

- **Correction interface with compliance control.** In order to apply delta correction over a running policy, we provide a compliant interface that allows humans to safely intervene and apply force to the robot to affect its behaviors at any time, as shown in Fig. 2. We design a kinesthetic correction hardware setup with a detachable handle for human to hold when correcting, and allows easy tool-swapping for different tasks. We run a compliance controller (specifically admittance control) in the background to respond to both contact forces and human correction forces, allowing the human to influence but not completely override the policy execution. The admittance controller uses a constant stiffness $\sim 1000$ N/m to allow easy human intervention and ensure accurate tracking.

- **Correction recording with buttons and force sensor.** Our interface additionally includes an ATI 6-D force sensor to directly measure contact forces, and a single-key keyboard placed on the handle to record the exact timings of correction starts/ends. Both the policy's original commands and the human's delta corrections are recorded, along with force sensor readings during the interaction.

### 3.2 Compliant Residual Policy

Given the correction data, there are multiple ways to update the policy. Common practices include *retraining* the base policy from scratch with both initial data and correction data, and *finetuning* the

base policy with only the correction data. However, *retraining* is costly as it requires updating the entire base policy network from scratch with all the available data. It also requires access to the base policy's initial training data, which might not be accessible for many open source pretrained models. The amount of correction data is significantly smaller than the initial training data, thus simply mixing them together makes the policy hard to gain effective corrective behaviors. While *finetuning* allows updating partial policy network parameters with new data only, its training stability can be easily affected by the distribution mismatch between the correction data and initial training data. Moreover, both retraining and finetuning can only update the policy with its fixed network architecture while being unable to incorporate new inputs and outputs. We propose a compliant residual policy trained only on the correction data to refine base policy's position actions and predict additional force actions.

**Compliant residual policy formulation.** Our policy directly learns corrective behavior from the human delta correction data, as shown in Fig. 3. It takes as input the same visual and proprioceptive feedback as the base policy but with a shorter horizon. It also takes in an extra force modality, which is available using our Compliant Intervention Interface. The policy outputs five frames of actions at a time, corresponding to $0.1\,s$ of execution time when running at $50\,Hz$. The action space is 15-dimensional: the first nine dimensions represent the SE3 delta pose from the base policy action to the robot pose command [8], while the later six dimensions represent the expected wrench (force and torque) the robot should feel from external contacts. Both the robot pose command and the expected wrench are sent to a standard admittance controller for execution with compliance.

The residual policy directly uses the base policy's frozen image encoder [35, 1, 46] to extract an image embedding, a temporal convolution network [39] to encode the force vectors, followed by fully-connected layers to decode actions.

**Advantages.** This formulation provides the following advantages:

- *Sample-efficient learning.* The residual policy's network is light-weight ($\sim$2MB trainable weights) and only requires a small amount of correction data to train (50$\sim$100 demonstrations).
- *Incorporating new sensor modality.* Compared to retraining and finetuning methods that are limited to the base policy's network architecture, residual policy can incorporate new sensor modality. This allows taking any position-based pretrained policy and turning it force-aware simply by collecting a small amount of correction data with force modality.
- *High-frequency inference.* The light-weight residual policy runs at a higher frequency than the base policy, incorporating high-frequency force feedback and enabling reactive corrective behaviors. This reactivity is particular important for error correction during contact events.

**Training strategy.** In prior work, a residual policy is trained either in simulation with RL [2, 51] to give it sufficient coverage of the state distribution, or in the real world with pre-collected behavior cloning data [31]. In this work, we train the Compliant Residual Policy completely on the small amount of new real-world correction data with the following strategies:

- *Ensure sufficient coverage of in-distribution data.* Human correction tends to be frequent around a few key moments of the task. A residual trained on correction data alone can extrapolate badly around states where no correction is provided. To help the residual policy understand when *not* to provide corrections, we: (1) include the no correction data for training but label it as zero residual actions; (2) collect a few trajectories where the demonstrator always holds the handle and marks the whole trajectory as correction even when the correction is small or zero. Details are in § A.3.
- *Prioritize correction data over no-correction (zero residual action) data.* Similar to [29], we alter the data sample frequency during training based on whether they have human correction or not. Specifically, since the moment of correction start indicates where the current policy performs badly followed by immediate action to fix it, we sample data more frequently for a short period immediately after correction starts. Our real-world ablations (§ 4.5) demonstrate that our training strategies improve the quality of the residual policy and the overall success rate.

## 4    Evaluation

For each task, we train a diffusion policy [8] with 150$\sim$400 demonstrations as the base policy. We first deploy the base policy and observe its performance and failure modes. Next, from the same base policy, we collect 50$\sim$100 correction episodes with each data collection method. Then, we update the policy using each network updating method and training procedure. Finally, we deploy the updated policies and evaluate their performance under the same test cases. Details of tasks and comparisons are described below.

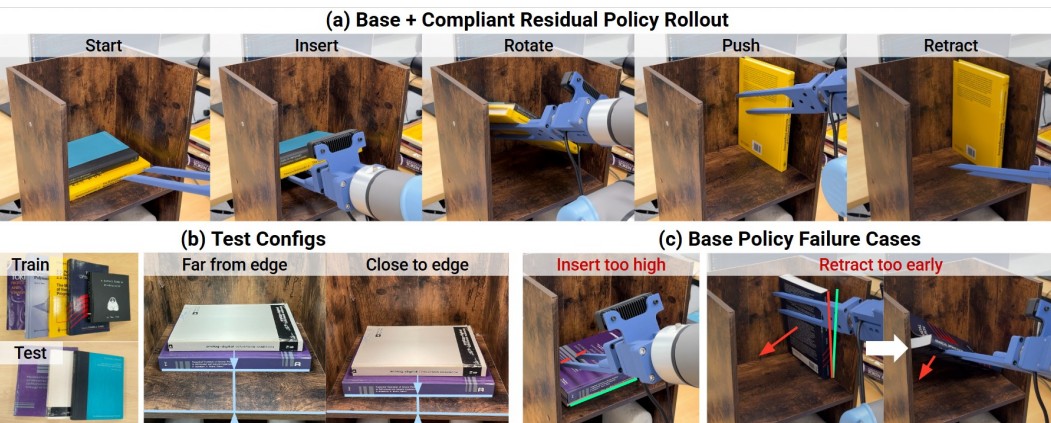

Figure 4: **Book Flipping Task.** (a) Policy rollout of [Compliant Residual] policy trained with [On-Policy Delta] data, demonstrating accurate insertion motions and forceful pushing strategy. (b) Different test scenarios. (c) Typical failure cases of the base policy: inserting too high above the book and missing the gap; retracting the fingers before the books can steadily stand.

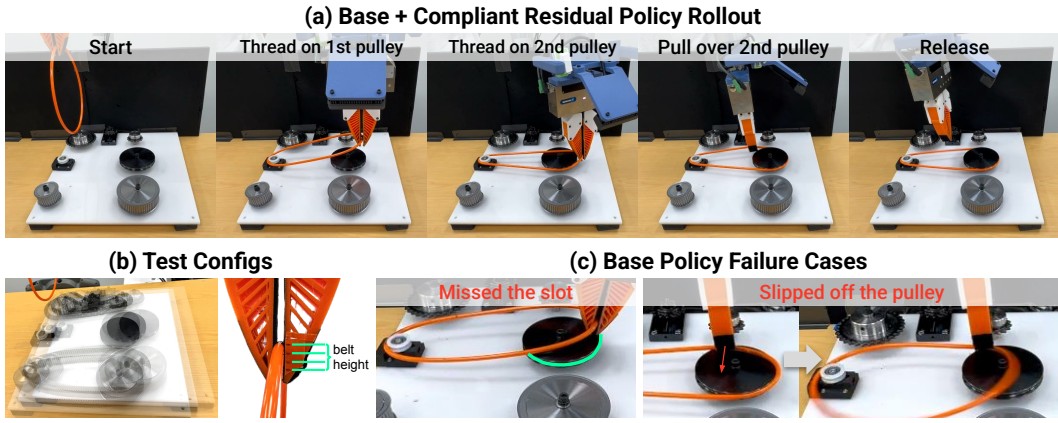

Figure 5: **Belt Assembly Task.** (a) Policy rollout of [Compliant Residual] policy trained with [On-Policy Delta] data, demonstrating accurate force-position coordination and adaptation. (b) Different test scenarios. (c) Typical failure cases of the base policy: missing the slot by going too high above the pulley; tilting the belt and causing it to slip off the pulley.

## 4.1 Contact-Rich Manipulation Tasks

**Book Flipping:** As shown in Fig. 4 (a), this task is to flip up books on a shelf. Starting with one or more books lying flat on the shelf, the robot first insert fingers below the book, then rotate the book up and push them firmly against the shelf to let them stand on their own. The base policy is trained with 150 demonstrations.

This task is challenging as it involves rich use of physical contacts and forceful strategies [20]. A position-only strategy always fails immediately by triggering large forces, so we execute all policies through the same admittance controller. The task success requires high precision in both motion and force to accurately align the fingers with the gap upon insertion, and to provide enough force to rotate heavy books and make the books stand firmly.

Each evaluation includes 20 rollouts on 4 test cases (5 rollouts each), as shown in Fig. 4 (b): 1) flipping a single book (three seen and two unseen books), initially far from the shelf edge; 2) flipping a single book close to the shelf edge; 3) flipping two books together (combinations of three seen and three unseen books), initially far from the shelf edge; 4) flipping two books close to the shelf edge. We use the same initial configurations for all evaluations.

**Belt Assembly:** As shown in Fig. 5 (a), this task is to assemble a thin belt onto two pulleys, which is part of the NIST board assembly challenge [26]. Starting with the belt grasped by the gripper, the robot needs to first thread the belt over the small pulley, next move down while stretching the belt to

thread its other side on the big pulley, then rotate 180° around the big pulley to tighten the belt, and finally pull up to release the belt from the gripper. The base policy is trained with 405 demonstrations.

The task is challenging as it requires both position and force accuracy throughout the process. Specifically, the belt is thin and soft so the initial alignments onto the pulleys are visually ambiguous. Also, since the belt is not stretchable, there is more resistance force and less position tolerance as the belt approaches the second pulley, requiring a policy with good force-position coordination and adaptation. We ran 32 rollouts across four different initial board positions and four grasp locations (Fig. 5 (b)).

## 4.2 Base Policy and its Failure Modes

We trained a diffusion policy [8] that takes in past images from a wrist-mounted camera and robot proprioception observations, and predicts a future position-based action trajectory. To isolate the contribution of force inputs versus human corrections, we trained diffusion policies with and without force inputs as baselines for the belt assembly task.

The book flipping base policy achieves a 40% success rate with the following common failure cases (Fig. 4 (c)): (1) Missed insertion. The fingers initially go too high above the book or aims for the gap between the two books, failing to properly insert beneath the books. (2) Incomplete flipping. At the last stage, the policy retracts the blade before the book can stand stably, causing it to fall back.

The belt assembly base policy achieves a 15.6% success rate. Adding force input increases the base policy success rate to 43.8%. Common failure cases include (Fig. 5 (c)): (1) Missed slotting: the fingertip goes too high or too low, causing the belt to miss the slot on the big pulley. (2) Belt slippage: the fingers pull the belt in the wrong direction, causing the belt to tilt and slip off the pulley.

## 4.3 Comparisons

We compare CR-DAgger with baselines across two dimensions: correction method and policy update method. We present the quantitative results in Fig. 6, and explain key findings in § 4.4.

**Correction data collection methods.** We compare our Compliant Intervention Interface with the two most commonly used correction data collection strategies:

- *Observe-then-Collect* includes two steps: first, the policy is deployed and human demonstrators observe the initial settings that could cause failures; then, demonstrators provide completely new demonstrations starting from similar initial settings. As explained in § 3.1, this type of offline correction potentially misses critical timing information, and the resulting demonstrations may deviate from the policy's original behavior distribution.

- *Take-over-Correction* (HG-DAgger) [23] is another common correction strategy where human demonstrators monitor policy execution and take complete control when failures are anticipated. We implement Take-over-Correction on our Compliant Intervention Interface by cleaning up command buffer to the compliance controller and switching stiffness to zero upon correction starts, so the robot policy does not affect the robot during correction. However, as explained in § 3.1, take-over correction introduces an abrupt transition around control authority switching, which may cause distributional discontinuities in the training data.

- *On-Policy Delta (Ours):* the details are described in § 3.1.

**Policy update methods.** We compare with two common policy update methods:

- *Retrain Policy:* Retrain the base policy using both the original training data and the correction data from scratch. As explained in § 3.2, this approach is reliable but may require access to the orignal data and large amount of new data to work well.

- *Finetune Policy:* Finetune the base policy using only the correction data (freezing visual encoders). As explained in § 3.2, this approach can be sensitive to data quality and distribution shifts.

- *Fintune Policy with KL Regularization:* A recent method [11] that stabilizes finetuning training by encouraging the predicted action to be close to the training data distribution.

- *Residual Policy:* an ablation of our method where force is removed from both input and outputs.

- *Compliant Residual Policy (Ours):* Residual policy update with additional force input and outputs, see details in § 3.2.

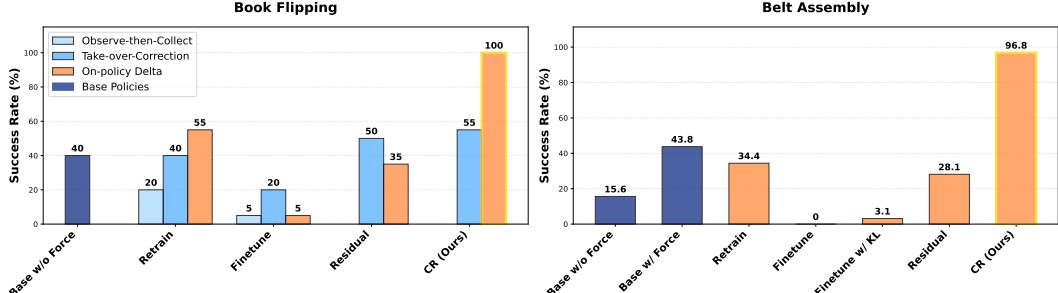

Figure 6: **Results.** We compare CR-DAgger across two dimensions: correction method and policy update method. The result shows that our [Compliant Residual (CR)] policy trained with [On-Policy Delta] data is able to improve upon base policies on both tasks and outperforms other variations.

## 4.4 Key Findings

**Finding 1: Compliant Residual Policy is able to improve base policy by a large margin.** As shown in Fig. 6, [Compliant Residual] policy trained with [On-Policy Delta] data improves the base policy success rate by 60% and 50% on the two tasks respectively. It effectively learns useful corrective strategies from the limited demonstrations. For example, in the book flipping task, the policy learns to pitch the fingers down more before finger insertion to increase the insertion success; in the belt assembly task, the policy learns to correct the height of the belt when misaligning to the pulley slot. Results are best viewed in our supplementary video.

**Finding 2: Residual policy allows additional useful modality to be added during correction.** [Compliant Residual] policy performs significantly better than other methods without force (45% higher success rate than the best position-only baseline on the book task and 53% higher on the belt task) as it can both take in force feedback that indicates critical task information and predict adequate contact forces to apply. For example, the last stage of the book flipping task requires the robot to firmly push the book against the shelf wall to let it stand on its own. [Compliant Residual] policy predicts large pushing forces at this stage to make the books stand stably with a 100% success rate, while [Residual]'s success rate drops from 70% to 35% (§ A.2). The second stage of the belt assembly task (threading the belt on the large pulley) requires delicate belt height adjustments under ambiguous visual information due to occlusions and the lack of depth. [Compliant Residual] policy learns to move along the pulley to find the slot when the finger touches the top of the pulley.

**Finding 3: Smooth On-Policy Delta data enables stable residual policy.** [Compliant Residual] policy has 45% higher success rate when trained on [On-Policy Delta] data instead of [Take-over-Correction] data on the book flipping task. Residual policy trained with [Take-over-Correction] data sometimes exhibits large noisy motions that trigger task failures, such as retracting the fingers too early in the book flipping task. On the contrary, residual policy trained with [On-policy Delta] data have much smoother action trajectories and better reflect human's correction intentions, providing suitable magnitudes of corrections.

**Finding 4: Retraining base policy is stable but learns correction behavior slowly.** Retraining from scratch with the initial and correction data together leads to policies with stable motions. However, its behavior is less affected by the small amount of correction data compared to the dominant portion of initial data, leading to insignificant improvements over the base policy (1.67% success rate drop on the book task averaged across all data collection methods, 18.8% success rate improve on the belt task, both are much less improvements than [Compliant Residual]).

**Finding 5: Finetuning base policy is unstable.** Policy finetuning with either correction data has the worst performance across all policy update methods and even underperforms the base policy (30% success rate drop on the book task averaged across all data collection methods, 15.6% drop on the belt task). The finetuned policy predicts unstable and noisy motions, quickly leading to out-of-distribution states, such as inserting too high in the book flipping task and drifting away from the board in the belt assembly task. This is likely due to the distribution mismatch between the base policy training data and correction data, causing training instabilities. Adding KL-regularization effectively reduced the noisy behavior, however, the overall success rate is still lower than other baselines.

## 4.5 Ablations

We study two important design decisions with ablation studies on the book flipping task.

**Training frequency and batch size.** One important parameter in DAgger is the batch size between policy updates. With a smaller batch size, the policy is updated more frequently, then new correction data can better reflect the most recent policy behavior. However, DAgger with small batch sizes is known to suffer from *catastrophic forgetting* [27, 12] since it finetunes neural networks on data with non-stationary distribution. Common solutions include retraining the residual policy at the end of DAgger using all available correction data collected from all the intermediate residual policies [41]. Another way is to rely on the base policy training data as a normalizer [17]. In this work, we empirically found that larger batch sizes can effectively stabilize residual training. With batch size = 50, the book flipping task reach 100% with one batch, while the belt assembly task performs better with two batches. We compare our method with a smaller batch size on the book flipping task, where we warm up the residual with 20 episodes of initial correction data, then update every ten more episodes for three times.

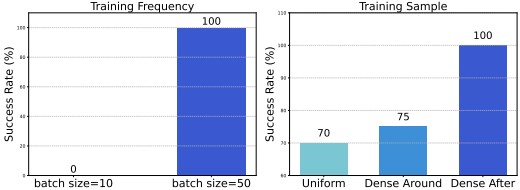

*Finding: Large-batch DAgger is more suitable for training Compliant Residual Policy.* The small-batch training becomes unstable and the demonstrator needs to provide large magnitudes of corrections as the number of iterations increases. During evaluation, the final policy always fails by inserting too high, while our single-batch policy achieves a 100% success rate with the same amount of data and training epochs.

Figure 7: **Effect of Training Frequency and Sample.** Batch size=50 leads to more stable training and dense sampling after correction starts achieves better performance on the book flipping task.

**Sampling strategy during training.** The start of a human intervention contains critical information of the timing and direction of correction. Accurate delta action predictions right after correction starts are important for reactive corrective behaviors. We investigate three strategies for sampling from online correction data during training: 1. Uniform sample, where the whole episode is sampled uniformly. 2. Denser sample around the start of a human intervention, and 3. denser sample only after the human intervention starts. For 2 and 3, we uniformly increase the sample frequency four times for a fixed period before and/or after intervention starts.

*Finding: Sampling denser right after intervention starts leads to more reactive and accurate corrections.* As shown in Fig. 7 (right), the best performance comes from densely sampling after the beginning of interventions. Sampling denser around the start of a human intervention also adds more samples right before the intervention starts, which is where humans observe signs of failures. These are mostly negative data, and using them for training decreases the policy success rate.

# 5   Conclusion and Discussion

In this work, we evaluate practical design choices for DAgger in real-world robot learning, and provide a system, CR-DAgger, to effectively collect human correction data with a Compliant Intervention Interface and improve the base policy with a Compliant Residual Policy. We demonstrate the effectiveness of our designs by comparing them with a variety of alternatives on four contact-rich manipulation tasks.

**Limitations and Future Work.**

The base policy should have a reasonable success rate for the residual policy to learn effectively. From our experiments, we recommend starting to collect correction data for the residual policy when the base policy has at least 10% ∼ 20% success rate. A future direction is to derive theoretical guidelines for the trade-off between the base and residual improvements.

Throughout this work, we use a MLP as the action head of our Compliant Residual Policy and directly regress the actions. Although it works well in our tasks, it may experience difficulty for tasks that involve more distinctive action multi-modalities. More expressive policy formulations, such as Flow Matching [28, 6] might be useful for these tasks.

Our data collection system is based on kinesthetic teaching. Although it provides richer data with higher quality than teleoperation as explained in the paper, it may require more labor during training data collection since the demonstrator needs to grasp the handle on the robot.

# 6    Acknowledgment

We would like to thank Chendong Xin for his support on experiments and rebuttal. We would like to also thank Eric Cousineau, Huy Ha, and Benjamin Burchfiel for thoughtful discussions on the proposed method, thank Mandi Zhao, Maximillian Du, Calvin Luo, Mengda Xu, and all REALab members for their suggestions on the experiment setup and the manuscript. This work was supported in part by the NSF Award #2143601, #2037101, and #2132519, Sloan Fellowship, Toyota Research Institute, and Samsung. We would like to thank Google and TRI for the UR5 robot hardware. The views and conclusions contained herein are those of the authors and should not be interpreted as necessarily representing the official policies, either expressed or implied, of the sponsors.

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

# A Technical Appendices and Supplementary Material

## A.1 Control Implementation Details

### A.1.1 Compliance Control

Compliance is a motor property that describes how motion responds to force. For example, a balloon has high compliance, meaning that they deform significantly under external forces. Industrial robots typically have very low compliance, meaning that their motion tracking has little deviation under external forces.

Compliance control refers to the technique that makes a rigid robot behave softly using feedback control. It is a standard technique widely adopted in the manufacturing industry. It lets robots interact with the environment safely without creating huge forces. With compliance, the robot retreats when it experiences external forces, larger force leads to bigger position deviations. When there is no external force, compliance control tracks the target position accurately just like position-based control. Compliance control allows specification of the exact desired compliance profile, typically described by parameters such as stiffness, damping and inertia.

Impedance control and admittance control are two methods to implement compliance control. The users are free to choose the controller that works with their robot. Robots with good position control accuracy (e.g., most industrial robots) can use admittance control. Robots with low gear ratio (e.g. "quasi-direct-drive" robots) can use impedance control. We use admittance control in this work and open-sourced our implementation at https://github.com/yifan-hou/force_control. A good reference for different compliance control schemes is the "Force Control" chapter of the Handbook of Robotics [40].

### A.1.2 Control Architecture

Our software system consists of three independent loops:

1. **The base policy loop** that runs the diffusion policy. The base policy loop updates at about 1Hz, each time predicts 32 frames of actions corresponding to 3.2s of future robot positions. In this work we have not optimized the implementation for computation speed.

2. **The residual policy loop** that runs at approximately 50Hz, each time predicts five frames of delta actions corresponding to 0.1s of delta positions. The delta actions are added to the corresponding base policy actions based one time, before being sent to the low-level compliance controller for execution. The loop rate is limited by our current implementation and can be improved if needed.

3. **The admittance controller loop** that runs at exactly 500Hz. This loop implements 6D Cartesian compliance on the robot. It takes reference positions and forces from the residual policy. When there is zero reference force and no external force, the admittance controller lets the robot track the reference position precisely. When external force exists, the robot will deviate from the reference position like a spring centered on the base policy output position.

Apart from the above controller/policy loops, each hardware (e.g. camera, force-torque sensor) has a standalone driver loop maintaining 1. communication with the hardware, and 2. buffers for action and feedback for this hardware.

## A.2 Stage-Wise Success Rate

We report the success rate of the book flipping task into three key stages.

## A.3 Correction Data Decomposition

As mentioned in the "Training strategy" part of § 3.2, we used two strategies to ensure the residual policy behaves stably around low correction data regions. The first strategy is to include the no correction portion of online data for training and label them with zero residual actions. The second strategy is to collect a few trajectories (15 out of the 50 total correction episodes) in which the

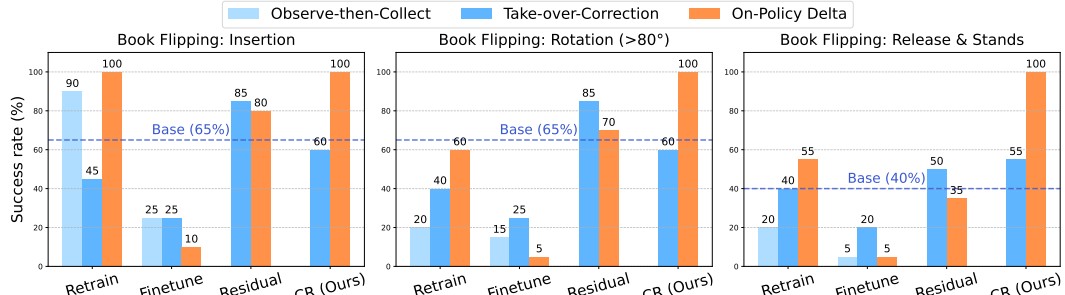

Figure 8: **Stage-wise Success rates.** Each row represents the results for one task, while each column shows the success rate up to the corresponding stage.

demonstrator marks the whole trajectory as correction, even when the correction is small or zero. In practice, we find that the first strategy works better when the base policy is more stable and has a higher success rate, while the second strategy works better otherwise. In our experiments, we use the first strategy for the book flipping task and use the second strategy for the belt assembly task.

### A.4 Experiments Compute Resources

We use a desktop with a NVIDIA GeForce RTX 4090 GPU for training and deployment.

### A.5 Hardware Design

Our kinesthetic correction hardware setup features a tool interface that allows task-specific tool swapping. For the book flipping task, we designed a customized fork-shaped tool that can easily insert under the books and flip them. For the belt assembly task, we used a standard WSG-50 gripper and fin-ray fingers [9]. An interesting future direction is to leverage generative models for automatic manipulator design [47]. Future work can also incorporate other types of force or tactile sensors, such as capacitive F/T sensors [10] and vision-based tactile sensors [50, 30, 4].

### A.6 Broader Impact

CR-DAgger contributes to the field of robotics by improving pretrained real-world manipulation policies with a small amount of human correction data. The proposed Compliant Intervention Interface provides an intuitive and safe way for humans to directly interact with robots and correct the robot policy on the spot. We demonstrate significant policy improvements on two real-world contact-rich manipulation tasks, book flipping and belt assembly, which can lead to useful applications in industry and households.

