# OpenReview forum: "Compliant Residual DAgger: Improving Real-World Contact-Rich Manipulation with Human Corrections"
_NeurIPS.cc/2025/Conference — NeurIPS 2025 poster_

### Official Review · Reviewer_UqHu · 2025-06-16

**Clarity:** 3
**Significance:** 2
**Originality:** 3
**Rating:** 4
**Confidence:** 2

**Summary:**

The authors propose CR-DAgger, a novel approach for improving robot manipulation policies in real-world contact-rich tasks. The key innovations include (1) a Compliant Intervention Interface, which is a kinesthetic, force-sensitive correction setup allowing human demonstrators to apply residual actions during policy execution via admittance control; (2) a Compliant Residual Policy, which is a lightweight residual policy trained only on correction data that augments a pretrained policy. It incorporates additional force feedback to generate corrective pose deltas and desired forces. The method is evaluated on two real-world manipulation tasks: booking flipping and belt assembly. Extensive experiments show the method significantly outperforms standard approaches like retraining and finetuning in terms of success rate and data efficiency.

**Questions:**

- Could the authors provide a more formal description of the learning paradigm for the Compliant Residual Policy? For instance, what is the exact loss function used? How are forces and pose predictions weighted during training?
- Have the authors tested CR-DAgger in more unstructured unseen cases? For example, more variations on the size/CoM location of the books or switching the locations of the small and large pulleys instead of minor changes on camera poses.

**Ethical Concerns:**

["NO or VERY MINOR ethics concerns only"]

**Final Justification:**

In the initial review, my main concerns are: 1. lack of technical rigor; 2. lack of evidence of generalization. The authors propose some revisions to address the lack of technical rigor in the rebuttal and promise additional experiments to show evidence of generalization. Given these, I lean towards acceptance, but I am not confident that all proposed revisions can be fully integrated, considering the amount of work involved. I have also adjusted my confidence level to reflect this.

**Limitations:**

The authors acknowledge limitations in section 5.

**Paper Formatting Concerns:**

See weaknesses

**Quality:**

2

**Strengths And Weaknesses:**

Strengths:
- [Quality]: The system is well-engineered and experimentally validated in realistic scenarios. Real-world experiments with contact-rich tasks make this work practically meaningful. On-policy correction and residual learning are thoughtfully designed to overcome distributional mismatch issues in DAgger-based learning.
- [Clarity]: The writing is generally clear and well-structured. The figures effectively convey the main concepts and results. The supplementary video help the reader understand the significance of each design decision and the empirical performance.
- [Significance]: CR-DAgger improves upon base policies with very little data (~50 episodes), which is valuable for robotics applications.
- [Originality]: The idea of corrections during policy execution combined with compliance control is natural. Adding force feedback as an additional input modality into a residual policy is well motivated.

Weaknesses:
- [Lack of Theoretical Depth]: While the system demonstrates promising empirical results, the manuscript lacks theoretical analysis or general principles that could guide its application to other domains, especially considering that NeurIPS is a learning conference. There is no formal characterization of when and why the proposed framework is expected to work. The learning formulation is only described in paragraph form with no mathematical formulations, which makes it difficult to understand the precise mechanics of the training procedure or how the loss functions are designed and optimized. The absence of technical rigor limits the work's contribution from a foundational standpoint.
- [Limited Generalization Evidence]: The experimental validation is restricted to two tightly controlled tasks with well-defined contact conditions. From Figure 4 and 5, the evaluation settings appear closely matched to the training configurations, and there is little evidence that the learned policies generalize to more unstructured variations. The lack of variety in test setups makes it hard to asses the robustness of the method, especially in settings with novel object geometries or out-of-distribution visual inputs.
- [Incremental Novelty]: The proposed system is a combination of well-established ideas: DAgger, residual learning, and compliance control. While the integration is thoughtful and practically valuable, the method's novelty lies primarily in system engineering rather than algorithmic innovations.
- [Structural Clarity]: The paper suffers from a somewhat scattered presentation. Specifically, key motivations are mixed into multiple sections rather than being clearly motivated up front. This makes the manuscript lack of narrative cohesion.

---

> ### Author Rebuttal · Authors · 2025-07-31
>
> > Q1: [Lack of Theoretical Depth]: lacks theoretical analysis or general principles. No formal characterization of when and why the proposed framework is expected to work. No mathematical formulations... mechanics of the training procedure or how the loss functions are designed and optimized.
>
> **Mathematical formulation**
>
> The base policy takes in images $I_t$ and proprioceptions $P_t$, and predicts $T_a$ frames of end-effector poses $A_t^p=(A_{t_0}^p, A_{t_1}^p, ..., A_{T_a}^p)$.
>
> The Compliant Residual Policy takes in images $I_t$, proprioceptions $P_t$, and forces $F_t$, and predicts $T_r$ frames of delta poses $\Delta A_t^p=(\Delta A_{t_0}^p, \Delta A_{t_0+1}^p, ..., \Delta A_{t_0+{T_r}}^p)$ and target forces $A_t^f$.
>
> The combined poses of $A_t^p$ and $\Delta A_t^p$, and target forces $A_t^f$ are taken by an admittance controller to command the robot.
>
> The admittance controller reads the force feedback $f$ then compute a position command $p$, such that the force-position relationship behave like a spring-mass-damper system:
> $$f = m \ddot p - \sigma \dot p + (p - p_0)k$$
> Where $m$, $\sigma$, $k$ are the inertia, damping, and stiffness parameters, respectively. $p_0$ is the resting position of the virtual spring. During policy rollout, $p_0$ is set to the base policy output position.
>
> **Training procedure and loss functions**
>
> The base policy is a UNet-based diffusion policy [1] that approximate the conditional distribution $p(A_t^p | I_t, P_t)$. It is formulated as a Denoising Diffusion Probabilistic Model (DDPM) [2], where the action generation is modeled as a denoising process (Stochastic Langevin Dynamics [3]). Starting from $A_t^{p, K}$ sampled from Gaussian noise, the DDPM performs K iterations of denoising $A_t^{p, K}, A_t^{p, K-1}, …, A_t^{p, 0}$, until a desired noise free final action $A_t^{p, 0}$. The process follows the equation:
> $$A_t^{p, k-1}=\alpha(A_t^{p, k}-\gamma\epsilon_\delta(I_t, P_t,A_t^{p, k}, k)+\mathcal{N}(0,\sigma^2 I) )$$
> Where $\epsilon_\delta$ is the noise prediction network that is optimized through learning, $\mathcal{N}(0,\sigma^2 I)$ is Gaussian noise added at each iteration, and $\alpha, \gamma, \sigma$ are parameters determined by a noise scheduler [2].
>
> The training loss of the base policy is:
> $$\mathcal{L}=MSE(\epsilon^k, \epsilon_\delta(I_t, P_t,A_t^0+\epsilon^k, k))$$
> Which is optimized through gradient descent.
>
> We employ a MLP architecture for the Compliant Residual Policy, which is theoretically able to capture the full correction demonstration distribution, supported by the Universal Approximation Theorem [4] - a MLP with a single hidden layer can approximate any continuous function. We use a temporal encoding via casual convolution [5] as the force encoder, which helps capturing causal relations from sequential data like force [6].
>
> Its training loss is:
> $$\mathcal{L}=MSE([\Delta A_t^p, A_t^f], \pi_{res}(I_t, P_t, F_t))$$
> Which is optimized through gradient descent.
>
> We will add these to the paper.
>
> **When and why the framework is expected to work**
>
> Our Compliant Residual Policy formulation allows additional modality being added to the policy.  Therefore, for contact-rich tasks where force is useful but may not be available in initial pretraining data, CR-DAgger is expected to work (Sec 4.4 Finding 2).
> Also, when there are clear failure modes from the base policy, and human correction data can fully represent corrective behaviors with vision, position, and force, CR-DAgger can effectively learn such strategies (Sec 4.4 Finding 1).
>
> [1] Chi, C., Xu, Z., Feng, S., Cousineau, E., Du, Y., Burchfiel, B., ... & Song, S. (2023). Diffusion policy: Visuomotor policy learning via action diffusion. The International Journal of Robotics Research.
>
> [2] Ho J, Jain A and Abbeel P (2020) Denoising diffusion probabilistic models. arXiv preprint arXiv:2006.11239.
>
> [3] Welling M and Teh YW (2011) Bayesian learning via stochastic gradient langevin dynamics. In: Proceedings of the 28th international conference on machine learning.
>
> [4] Cybenko, G. (1989). Approximation by superpositions of a sigmoidal function. Mathematics of control, signals and systems.
>
> [5] Aaron van den Oord. “WaveNet: A Generative Model for Raw Audio”. In: arXiv preprint arXiv:1609.03499 (2016).
>
> [6] Hou, Y., Liu, Z., Chi, C., Cousineau, E., Kuppuswamy, N., Feng, S., ... & Song, S. (2024). Adaptive compliance policy: Learning approximate compliance for diffusion guided control. arXiv preprint arXiv:2410.09309.
>
> > Q2: [Limited Generalization Evidence]: The experimental validation is restricted to two tightly controlled tasks with well-defined contact conditions... there is little evidence that the learned policies generalize to more unstructured variations.
>
> The book flipping task is evaluated with both known and novel books, as shown in Figure 4 (b). The test books differ significantly with train books in terms of size, weight, and appearance. We plan to also add another task of cable routing where we will test with several novel cables with OOD visual appearance, length, and stiffness.
>
> Also, rather than having “well-defined contact conditions”, **we specifically choose tasks that  involve complex, multi-modal contact modes**. In the book flipping task, the initial finger insertion may touch the table in front of the book when there is a large enough gap, or it has to touch the front ridge of the book and push it back a little before touching down on the table. The insertion motion may or may not involve a contact between the fingertip and the back of the shelf wall, depending on the weight of the book and the policy strategy. The flipping motion after the rotation may simply be a pivoting of the book against the right shelf wall when the book is close to the right shelf wall; otherwise it must push the book to slide to the right, during which the book could slide off the fingers. In the belt task, the belt could get stuck on the base of the pulley or nearby gears/chains on the board when approaching the first pulley. The belt also may not move perfectly into the second pulley, instead, the following could happen: the finger may move down too early and hit the top of the pulley first; the finger may move too far away and stretch the belt too much so the belt ends up on the top of the pulley; the finger may move down too much and move the belt below the pulley slot. The loose side of the belt could get stuck on the wrong side of the pulley button.
>
> We already showed the complexity of the tasks from the list of failure modes in the experiment section (Fig.4 and Fig.5). We will include a more complete list of failure cases and explanations in the final paper.
>
> > Q3: [Incremental Novelty]: The proposed system is a combination of well-established ideas: DAgger, residual learning, and compliance control... the method's novelty lies primarily in system engineering rather than algorithmic innovations.
>
> DAgger, residual learning, and compliance control are very general research directions that are still actively being explored, rather than “well-established ideas”. There are many valuable works that continue to improve these formulations or leverage these techniques in a new way to solve important robotics problems. We believe that building on top of existing ideas and expanding existing research directions do not devalue a paper.
>
> Our technical contributions represent fundamental novelties, not just engineering combinations:
> - We propose a compliant residual interface that allows collecting on-policy delta correction data with haptic feedback, which ensures smoothness of correction data while maintaining consistency between correction data distribution and policy induced state-action distribution.
> - We propose a novel compliant residual policy formulation that effectively learns correction strategies for contact-rich tasks and enables any position-only base policy to become force-aware through minimal correction data. We also demonstrate a novel insight that on-policy delta corrections and force modality enable stable residual learning.
> - We are the first to demonstrate that DAgger can work effectively for contact-rich tasks with as few as 50 correction demonstrations, while prior works on DAgger often overlook and struggle with contact-rich tasks.
> - We present a systematic study of intervention interfaces and policy update methods.
>
> > Q4: [Structural Clarity]: The paper suffers from a somewhat scattered presentation. Specifically, key motivations are mixed into multiple sections rather than being clearly motivated up front.
>
> We present high-level key motivations in the introduction: “How to collect informative human correction data” and “How to effectively update the policy with new data”. Then in the method section, we motivate each of our system/algorithm design choices before introducing it. We appreciate any detailed suggestions from the reviewer regarding how specific sections and areas of the paper could be adjusted to improve the overall clarity of the manuscript.
>
> > Q5: Could the authors provide a more formal description of the learning paradigm for the Compliant Residual Policy? For instance, what is the exact loss function used? How are forces and pose predictions weighted during training?
>
> The formulation is answered in Q1.
> During training, the joint pose and force predictions are normalized across each channel dimension and weighted equally.
>
> > Q6: Test in more unstructured unseen cases? More variations on the size/CoM location of the books or switching the locations of the small and large pulleys instead of minor changes on camera poses.
>
> As explained in the answer to Q2, our experiment setup does involve variations of item/setup positions as well as novel items not seen during training.
>
> Regarding “minor changes on camera poses”, we would like to clarify that all our experiments use a consistent single wrist-mounted camera for policy observation, which moves dynamically with the robot end-effector.

---

> ### Comment · Reviewer_UqHu · 2025-08-05
> **Post-rebuttal**
>
> I appreciate the detailed response from the authors. Most of our concerns are resolved. However, I am still not convinced that the current stack of tasks is a diverse set or shows evidence for generalization, echoing Reviewer NN22 and J472. I understand I am not allowed to ask for more experiments during the Discussion phase. In this case, could the authors elaborate on the "task of cable routing" mentioned in the rebuttal? Specifically, **how** and **why** do the authors believe this additional task setting will add value to the comprehensiveness of the evaluations?

---

> ### Author Response · Authors · 2025-08-06
>
> Thank you for the thoughtful follow-up. The cable routing task involves threading a flexible cable through multiple mounting clips – a common and highly challenging manipulation task in real-world manufacturing. We plan to evaluate the policies on cables with varying stiffness, diameter, length, and visual appearance, all of which are not seen during training.
>
> This task expands the diversity and generalization scope of our evaluation for the following reasons:
> 1. Continuous deformable object manipulation: Unlike rigid books or semi-flexible belts, cables require reasoning over continuously deformable objects. The system must handle high-dimensional deformation states that evolve throughout the trajectory.
> 2. Sliding friction and sequential contacts: Cable routing involves making sequential contacts across multiple clips with sliding friction. This is different than the two-point pulley configuration in our belt task, which mainly involves stretching forces.
> 3. Tight space navigation: Routing through constrained, multi-fixture environments requires fine-grained force control and precise positioning to avoid cable kinking or jamming.
> 4. Broader generalization: Together with our book flipping experiments (which already demonstrate generalization to novel book sizes, appearances, and weights as shown in Fig. 4b), the cable routing task tests the policy's ability to handle different contact dynamics, material properties, and geometric configurations.

---

> > ### Comment · Reviewer_UqHu · 2025-08-07
> > **Post-Rebuttal Discussion**
> >
> > Thank you for the detailed clarification. With this extension, I lean towards acceptance and will update my ratings accordingly. However, the authors propose substantial additions to both the experimental section and the core method description. I am concerned about whether all of them can be fully integrated and polished in time for camera-ready.

---

> > > ### Author Response · Authors · 2025-08-09
> > >
> > > Thank you for your acknowledgement and for updating your rating. We are confident that we can complete the proposed updates within the camera-ready deadline.
> > > 1. Regarding the cable routing experiments: We have already finished setting up the experiment and tested our method on in distribution scenarios. The base policy achieves a 90% success rate, while our Compliant Residual Policy reaches 100% success. We are currently working on adding more challenging scene variations and novel items, and expect to finish all evaluations in 1-2 weeks. We have added these in-domain experimental results to our project website.
> > > 2. Regarding writing updates: Our core method remains unchanged—all proposed additions involve clarifications and backgrounds already drafted in our rebuttal responses. Specifically, we will integrate the mathematical formulations and compliance control background suggested by the reviewers. We will reorganize content between the main paper and appendix to meet page limits while maintaining clarity.
> > >
> > > We believe that we have sufficient time to polish our paper that fully addresses the reviewer feedback before camera ready.

---

### Official Review · Reviewer_NN22 · 2025-07-03

**Clarity:** 4
**Significance:** 3
**Originality:** 3
**Rating:** 5
**Confidence:** 4

**Summary:**

The paper introduces a residual policy to learn the human force correction during robot demonstration collection, which is verified to be effective in challenging contact-rich tasks over existing Dagger methods. This seems to be an efficient way to collect demonstrations, with an additional human handle and force sensor required.

**Questions:**

The base policy (pose) and residual policy (force) are learning two types of signals, how are these two combined into the admittance controller? Would there be both the impedance control and admittance control? Will these raise some requirements on robotic control systems that can be hard to work for all popular robots?

Finetuning base policy is unstable, how about adding additional regularization (e.g. KL) with base policy as reference?

**Ethical Concerns:**

["NO or VERY MINOR ethics concerns only"]

**Final Justification:**

Thanks for the rebuttal and additional results. I will keep my score.

**Limitations:**

yes

**Quality:**

3

**Strengths And Weaknesses:**

Strengths

The paper is well written and easy to follow.

The motivation is well explained, that conventional off-policy correction can suffer from distribution shifts and additional errors.

The compliant intervention interface and compliant residual policy is novel.

The alation on design choices like batch size and sampling strategy is insightful.

Base policy should be able to accomplish most tasks without requiring detailed force control, the lightweight residual policy can be applied for contact-rich tasks in an efficient manner.

Different training recipes are discussed with details revealed, which is insightful.



weakness

The experiments can be conducted on more diverse contact-rich tasks.

---

> ### Author Rebuttal · Authors · 2025-07-31
>
> > Q1: The experiments can be conducted on more diverse contact-rich tasks.
>
> We plan to add another experiment on a cable routing task. It involves threading a flexible cable through multiple clips. The task requires precise force control to avoid cable kinking while navigating tight spaces.
>
> > Q2: The base policy (pose) and residual policy (force) are learning two types of signals, how are these two combined into the admittance controller? Would there be both the impedance control and admittance control? Will these raise some requirements on robotic control systems that can be hard to work for all popular robots?
>
> The base policy, the residual policy and the compliance controller run in separate threads. They each take required input directly from the sensors. The base policy and residual policy outputs are combined into target position and forces, and are passed to the compliance controller as inputs.
>
> Impedance control and admittance control are two methods to implement compliance control. The users are free to choose the controller that works with their robot. Robots with good position control accuracy (e.g., all industrial robots like ABB, Fanuc, Kuka, UR, etc.) can use admittance control.  Robots with low gear ratio (e.g., Unitree, ARx) can use impedance control. In this work we choose admittance control because we use an UR robot.
>
> We will add a paragraph to the paper to explain our controller structure in more detail, and introduce compliance control, which is a standard technique widely adopted in the manufacturing industry.
>
> > Q3: Finetuning base policy is unstable, how about adding additional regularization (e.g. KL) with base policy as reference?
>
> We thank the reviewer for the suggestion. We added a new baseline that finetunes the base policy with KL regularization. Our implementation follows DPOK [1]. The results on the belt assembly task are listed below:
>
> | Method | Success Rate |
> |--------|--------------|
> | Base Policy | 15.625% |
> | Retrain | 34.375% |
> | Finetune | 0% |
> | Finetune w/ KL | 3.125% |
> | Residual Policy | 28.125% |
> | Compliant Residual Policy | 96.825% |
>
> Although [Finetune w/ KL] slightly improved the success rate over [Finetune] and exhibited smoother and stabler behaviors, it still hurts the [Base Policy] performance by 12%, while [Compliant Residual Policy] demonstrates a much more significant improvement over [Base Policy] by 81.2%. We have also added the rollout videos to the project website.
>
> [1] Fan, Y., Watkins, O., Du, Y., Liu, H., Ryu, M., Boutilier, C., ... & Lee, K. (2023). Dpok: Reinforcement learning for fine-tuning text-to-image diffusion models. Advances in Neural Information Processing Systems, 36, 79858-79885.

---

### Official Review · Reviewer_J472 · 2025-07-03

**Clarity:** 2
**Significance:** 4
**Originality:** 3
**Rating:** 4
**Confidence:** 5

**Summary:**

This paper tackles the challenge in collecting correction data for robot policies.  Prior works are mostly based on two approaches: (1) observe-the-collect, an offline method that re-collect successful trajectories after observing failed operation of robots, and (2) take-over-correction, an online method that introduce human's interference when failures are about to occur.  However, these approaches are limited, as the former fails to cover the the policy’s typical failure scenarios and the later introduces non-smooth transitions or errorneous correction due to spatial mismatch and teleoperationlatency.  This paper proposes an online alternative that leverages compliance control to provide smooth and instantaneous action corrections.  Moreover, this paper introduces a compliant residual policy that predicts residual actions and intended force control for the base policy.  Carrying out experiments on real-world manipulation tasks, this paper demonstrations significant performance gain over position-control baselines.

**Questions:**

1. What is the formulation of the compliance control?

2. What does "the policy’s original distribution" at line 107 mean? What does "distribution shift" mean at line 116? What does "input distribution" mean at line 184?

3. What are the evidence that support the statement of "distribution drift" and "distributional consisntency"?  (See Weakness 3 for more details)

4. Does the performance gain result from force feedback or force control during inference?

5. How to generate the ground-truth of the external force output for the residual policy?

6. Does the policy generalize better to novel scene configuration than policies using position-based control?

**Ethical Concerns:**

["NO or VERY MINOR ethics concerns only"]

**Final Justification:**

The rebuttal has addressed my concerns in writing.

**Limitations:**

Yes

**Quality:**

2

**Strengths And Weaknesses:**

Strengths:

1. This paper proposes a novel and interesting approach, that collects correction data with compliance control, on DAgger.  The system allows more accurate, gentle and instantaneous corrections to the base policy.

2. This paper clearly illustrates the categories and their corresponding limitations of prior arts.  Offline methods fail to cover failure scenarios, while online methods using position-based control introduct non-smooth behaviors in the correction data.  Reads can easily appreciate the idea of using compliance control for data collection.

3. The manipulation tasks considered in this paper involve more complex physical interaction than typical pick-and-place tasks, which necessitates the usage of compliance control.

---

Weaknesses:

1. This paper does not provide an explicit formulation of "compliance control".  Without solid background knowledge in control, I can only guess the action exectuion of robot policies is softer and smoother than position-based control.  However, I'm not sure if the system desires a small and constant external force during action execution or imposes an upper bound on the external force.

2. The authors abuse the term--"distribution", throughout the paper.  For example, the authors refer to "state distribution" at line 22, while referring to "action distribution" at line 26.  With multiple meanings of the same word, it becomes challenging to read and understand the paper.  For example, what does "the policy’s original distribution" at line 107 mean? what does "distribution shift" mean at line 116? what does "input distribution" mean at line 184?

3. The writing of this paper is often vague and imprecise.  For example:

    - The paragraph of **distribution shift** is inconclusive.  Since both smooth and non-smooth correction behaviors deviate from the original action distribution of the base policy, CR-DAgger does not resolve the challenge of distribution drift.  There are other reasons that make CR-DAgger more effective than position-based DAgger.  Describing the problem without addressing it is weird.
    - At line 40, the authors said "*Unlike take-over corrections that may cause force discontinuity, our design allows smooth transition between correction/no correction mode, while maintaining distributional consistency with the original policy.*".  However, I'm not sure what consistency is maintained by CR-DAgger.

    These statements are not supported by any analysis.

4. This paper does not ablate the effectiveness of force feedback.  Does the performance gain result from force feedback or force control during inference?

5. This paper lacks technical details in the generation of the ground-truth of external force output for the residual policy.  The force sensor only keep tracks of the total force exerted on the sensor.

6. This paper does not discuss the limitation that the human teleoperator is required to hold the teleoperation interface during the deployment of policies.  Such a teleoperation interface requires more labor effort than teleoperation via mouses or keyboards.

7. This paper does not discuss the generalization of the compliance-control residual policy to novel scenarios.  Conditioning on force feedback, does the policy generalize better to novel scene configuration than policies using position-based control?

8. I'm afraid this paper hinders the real power of compliance control.  Going beyond data correction, probably we should collect data with compliance control and train policies with force control, rather than those with position-based control, for imitation learning.

---

> ### Author Rebuttal · Authors · 2025-07-31
>
> > Q1: Explicit formulation of "compliance control"... I'm not sure if the system desires a small and constant external force during action execution or imposes an upper bound on the external force.
>
> We will add an introduction of compliance control to the paper. Compliance control is a traditional control technique to make a robot behave like a spring when experiencing external forces. It lets robots interact with the environment safely without creating huge forces. With compliance, the robot retreats when it experiences external forces, larger force leads to bigger position deviations. When there is no external force, compliance control tracks the target position exactly just like position-based control. So it does not require or “desire a certain type of external force”, and does not “impose bounds” on those forces. Rather, it responds continuously to the external force. Compliance control is a standard technique widely adopted in the manufacturing industry. The “admittance control” we used is one popular way to implement compliance control. A good reference is the “Force Control” chapter of the Handbook of Robotics [1].
>
> [1] Villani, Luigi, and Joris De Schutter. "Force control." Springer handbook of robotics. Cham: Springer International Publishing, 2016. 195-220.
>
> > Q2: The authors abuse the term--"distribution"... "state distribution" at line 22, while referring to "action distribution" at line 26... what does "the policy’s original distribution" at line 107 mean? what does "distribution shift" mean at line 116? what does "input distribution" mean at line 184?
>
> Those “distribution” all refer to “the complete state-action distribution generated by a policy”, where the policy can be different depending on the context. In the final submission, we will standardize our terminology as follows:
> - “Policy distribution” refers to the joint distribution of states and actions encountered when deploying a specific policy.
> - “Distribution shift” refers to the mismatch between the state-action distributions in base policy training data and correction data.
> - “Input/state distribution” refers to the distribution of observations (states) the policy receives.
>
> > Q3: The writing is often vague and imprecise.
> The paragraph of distribution shift is inconclusive...what consistency is maintained by CR-DAgger.
>
> The paragraphs of concern describe the improvements of data quality from using our Compliant Intervention Interface, compared with the traditional “take over” correction.
> We validated the improvements of data quality from our Compliant Intervention Interface by comparing policies trained with both our correction data [On-Policy Delta] and [Take-Over-Correction] data. Policies trained with our data generally perform better, as shown in Fig.6 and discussed in Section 4.4 Finding 3.
>
> We agree that there could be more direct analysis of the data quality improvements, including less distribution shift and force discontinuity. We will add more analysis in the revised paper:
>
> - **[On-Policy Delta] introduces less distribution shift.**  We added a figure on our website that shows the distribution of fingertip trajectories across all dimensions in base policy training data, [On-Policy Delta] and [Take-Over-Correction] data. It demonstrates that [On-Policy Delta] data’s distribution is better aligned with base policy training data’s distribution than [Take-Over-Correction] data.
> - **[On-Policy Delta] enables smoother trajectories.** We also added figures comparing velocity magnitude within 1.5s of the corrections starts/ends to our project website. [On-Policy Delta] velocity magnitudes are smaller and more consistent, [Take-Over Correction] has notably larger magnitude and variations, demonstrating that [On-Policy Delta] encourages smoother trajectories.
>
> To further clarify, compliant Intervention Interface allows humans to directly apply forces to the robot when the base policy is continuously running to correct its behavior. Firstly, humans are applying delta actions on top of base policy actions, instead of overwriting them. Additionally, the human demonstrator can always sense the policy’s intention through haptic feedback, and easily control the magnitude of delta actions by the amount of force applied to the handle. As a result, the interface limits humans from providing large corrections to the base policy, ensuring the state-action pairs encountered during correction are close to the state-action distribution generated by the base policy, thus maintaining distributional consistency with it.
>
> > Q4: Does not ablate the effectiveness of force feedback. Does the performance gain result from force feedback or force control during inference?
>
> There are a few ways to interpret the question:
>
> “Does the performance improvement come from adding force feedback or from human correction?”
>
> We already showed that adding force feedback significantly improves the residual policy (comparison between [Residual] and [Compliant Residual] in Fig. 6; Section 4.4 Finding 2).
> We also added another baseline [Base Policy with Force] which added force input to the base policy, and showed that our [Compliant Residual] policy improves [Base Policy] more effectively than [Base Policy with Force], as shown in the results below (on the belt assembly task). This demonstrates the contribution of human correction in addition to force.
>
> | Method | Success Rate |
> |--------|--------------|
> | Base Policy | 15.625% |
> | Base Policy with Force | 43.75% |
> | Residual Policy | 28.125% |
> | Compliant Residual Policy | 96.825% |
>
> “The Compliant Residual Policy takes force feedback, and outputs force command. Does the performance gain result from only one of them?”
>
> Both force feedback and force command are important components of our proposed method. We want to argue that both are useful and it does not provide much insight to ablate one of them. If our system does not have force feedback, then it is not possible to compute the force command label to train a policy with force command. If we do not use force command, then the position command alone cannot fully describe the demonstrator’s intention.
>
> “You use a force controller to run the experiments. Maybe the performance gain comes from this force controller, not your learned policy.”
>
> We use force control (admittance control) for all policies being evaluated, as position control would immediately trigger the emergency stop in these contact-rich tasks and break the object or robot due to the large force exerted.
>
> > Q5: The generation of the ground-truth of external force output for the residual policy. The force sensor only keep tracks of the total force exerted on the sensor.
>
> The total force exerted on the additional force sensor is exactly the external contact force as described in Section 3.1 “Correction recording with buttons and force sensor”, since the handle is mounted behind the sensor so human force would not affect its reading.
>
> > Q6: Limitation that the human teleoperator is required to hold the teleoperation interface during the deployment of policies... requires more labor effort than teleoperation via mouses or keyboards.
>
> We agree that our kinesthetic teaching style correction may require more labor than teleoperation-based correction. We will add this to the limitation section. However, we want to point out that this is not a major concern because:
> - Additional labor is only needed when collecting correction data. Once the residual policy is trained, the robot can execute the task independently with a high success rate. The handle is also detachable and can be easily removed before evaluation.
> - Kinesthetic teaching provides key benefits including allowing intuitive and gentle correction, recording of human intended contact forces, and direct haptic feedback for the demonstrator. None of these benefits are available in traditional teleoperation systems with mouses or keyboards.
>
> > Q7: The generalization of the compliant residual policy to novel scenarios.
>
> The book flipping task is evaluated with both known and novel books, as shown in Figure 4 (b). The test books differ significantly with train books in terms of size, weight, and appearance. We plan to also add another task of cable routing where we will test with several novel cables with OOD visual appearance, length, and stiffness.
>
> > Q8: This paper hinders the real power of compliance control. Going beyond data correction, we should collect data with compliance control and train policies with force control, rather than those with position-based control.
>
> That is a good point. In fact, we already used compliance control when collecting even the base policy data. Compliance control is the reason why we can drag the robot around (kinesthetic teaching) to do demonstrations.
>
> However, current robot datasets and pretrained policies lack force modality, while our approach offers an easy solution to make such position-only policies force-aware with a small amount of correction data. Moving forward, while we hope that more and more public robot datasets and pretrained policies can include the force modality by default, we highlight that our approach is still able to provide performance improvements in the initial absence of force information by integrating it successfully through a residual prediction.
>
> > Q9: Formulation of the compliance control?
>
> The compliance controller we use is an admittance controller, meaning that at every time step it reads the force feedback $f$ then compute a position command $p$, such that the force-position relationship behave like a spring-mass-damper system:
> $$f = m \ddot p - \sigma \dot p + (p - p_0)k$$
> Where $m$, $\sigma$, $k$ are the inertia, damping, and stiffness parameters, respectively. $p_0$ is the resting position of the virtual spring. During policy rollout, $p_0$ is set to the base policy output position. We will add a section in the appendix to briefly introduce compliance control.

---

> > ### Comment · Reviewer_J472 · 2025-08-04
> >
> > The authors have addressed my concerns, and I'm happy to increase my ratings to borderline accept.

---

### Official Review · Reviewer_YHb6 · 2025-07-07

**Clarity:** 4
**Significance:** 3
**Originality:** 3
**Rating:** 5
**Confidence:** 4

**Summary:**

This paper presents a data-efficient method for improving pretrained policies on contact-rich robot manipulation tasks - by collecting delta action correction data and force feedback from human, and using this data to train a force-aware residual policy. The data collection method employs admittance control for human operators to injecting necessary intervention online while the robot follows base policy actions (10Hz). The collected data is used to train a residual policy that predicts delta action and autoregressively predict expected force (50Hz), which operates on top of the base policy in deployment.

The paper conducts extensive experiments and studies on two real-world tasks. By comparing between various ablations and baselines, the authors show that the proposed strategy significantly outperform other existing strategies.

**Questions:**

- How is the frequency of the compliant residual policy to be selected? WIth executed without the residual policy, does the admittance control run at the same frequency as well?

**Ethical Concerns:**

["NO or VERY MINOR ethics concerns only"]

**Final Justification:**

I will keep my score.

**Limitations:**

Yes

**Quality:**

4

**Strengths And Weaknesses:**

## Strength

- I enjoy reading this paper for its clarity and transparency. The problem it studies is well motivated, and the proposed methods address the problem well, backed by sufficient experimental results.
- Incorporating force feedback for human correction and policy deployment in contact-rich tasks is a foundamental problem in robot manipulation. I'm glad to see the authors propose a viable solution that effectively improves the success rate.
- Many of the discussions are informative and insightful. In particular, I like how the method is compared with different correction data collection strategies, as well as different training methods.

## Weakness

- There are two aspects that may explain the performance gain introduced by the complaint residual policy on top of the base policy: one is the human correction, and another is the force modality. By reading the paper, it's not clear to me how these two aspects contribute to the success of complaint residual policy. It would be very helpful if the authors provide more comparison on this, for example, training the base policy with force input as well and compare its results with complaint residual policy.

---

> ### Author Rebuttal · Authors · 2025-07-31
>
> > Q1: There are two aspects that may explain the performance gain introduced by the complaint residual policy on top of the base policy: one is the human correction, and another is the force modality. By reading the paper, it's not clear to me how these two aspects contribute to the success of complaint residual policy. It would be very helpful if the authors provide more comparison on this, for example, training the base policy with force input as well and compare its results with complaint residual policy.
>
> We thank the reviewer for the great question.
> To isolate and evaluate the contribution of human correction, we added a baseline [Base Policy with Force], where we train a diffusion policy that also takes force input on the same set of demonstration data. As shown in the results below on the belt assembly task, while [Base Policy with Force] improves success rate over [Base Policy] by 28.125%, [Compliant Residual Policy]’s improvement is a much more significant 81.2%, showing the contribution of human corrections in addition to force modality. We’ve also added the rollout videos to the project website.
>
> | Method | Success Rate |
> |--------|--------------|
> | Base Policy | 15.625% |
> | Base Policy with Force | 43.75% |
> | Residual Policy | 28.125% |
> | Compliant Residual Policy | 96.825% |
>
> On the other hand, our initial submission already shows the contribution of force modality in addition to human correction: the [Residual Policy] baseline is the same as our [Compliant Residual Policy] except that force is removed from both its inputs and outputs. Our [Compliant Residual Policy] consistently outperforms these baselines, demonstrating that force modality is beneficial for learning correction behaviors of contact-rich manipulation tasks, as shown in Sec 4.4 Finding 2.
>
> Therefore, both human correction and force modality contribute to the success of the compliant residual policy - by learning effective correction behaviors, which are often indicated by force modality, the compliant residual policy consistently improves the initial base policy.
>
>
> > Q2: How is the frequency of the compliant residual policy to be selected? WIth executed without the residual policy, does the admittance control run at the same frequency as well?
>
> The 50Hz frequency of compliant residual policy is upper bounded by the computation time of our entire control loop. We have not optimized our implementation particularly for computational efficiency. This frequency is sufficient for table top manipulation tasks in our experiments.
>
> A separate standard admittance controller is running at low level at 500Hz to make the robot compliant while tracking the target position (and force, if given) from the policy. It runs in its own thread and uses the same parameters across all baselines.

---

> > ### Comment · Reviewer_YHb6 · 2025-08-05
> >
> > Thank you for the rebuttle and additional results. I will keep my score.

---

### Note · Authors · 2025-08-14

We sincerely thank all reviewers for their thoughtful feedback and constructive suggestions on our work, CR-DAgger. We are encouraged by the positive reception of our paper, with reviewers finding our problem formulation well-motivated, our compliant intervention interface novel, our compliant residual policy effective, and our experimental design and results impressive.

We addressed key feedback from the reviewers and provided detailed clarifications regarding:

- We evaluated CR-DAgger against **a more robust baseline [Finetune with KL Regularization]**, demonstrating consistent and significant performance improvements (3.125% → 96.825% success rate) that validate our method's effectiveness.
- We introduced the **[Base Policy with Force] baseline** to isolate the contributions of force modality versus human corrections, showing that while adding force alone improves performance (15.625% → 43.75% success rate), incorporating human corrections provides more significant gains (43.75% → 96.825% success rate).
- We are adding **a cable routing task with positive preliminary results**. This task involves continuous deformable object manipulation with different contact dynamics from our existing tasks, broadening the diversity and generalization scope of our evaluation.
- We **addressed the reviewers’ concerns on writings** by providing mathematical formulations for policy learning and background on compliance control, while emphasizing that our core method remains unchanged.

We are pleased that reviewers recognize the significance and practical value of our contributions and have found the clarifications and additional experiments provided during the rebuttal process to be satisfactory. We are confident in completing all promised evaluations and integrating these improvements into our final manuscript.

---

### Decision · Program_Chairs · 2025-09-17

**Decision:**

Accept (poster)

**Comment:**

This paper addresses Dataset Aggregation (DAgger) for real-world contact-rich manipulation, an important problem where traditional approaches struggle with force dynamics and physical interaction complexities. The method introduces a novel Compliant Intervention Interface using compliance control for smooth human corrections and a Compliant Residual Policy that integrates force feedback.

Reviewers praised the clear motivation, well-engineered system design, and impressive experimental results. Some found the compliant intervention interface novel and the significant performance gains on challenging real-world tasks compelling.

Reviewers raised concerns about lack of mathematical rigor, unclear terminology, insufficient generalization analysis, and the need to isolate contributions of force modality versus human corrections. Some questioned whether the work represented algorithmic innovation versus primarily system engineering.

The authors largely addressed these concerns in their rebuttal, providing mathematical formulations, adding key ablation studies, clarifying terminology, and proposing an experiment for broader generalization evaluation.

One concern that may not have been fully addressed is Reviewer J472's question regarding generalization comparison to baselines—the authors' rebuttal under Q7 primarily highlights what generalization their experiments require rather than providing comparative evidence.

Finally, the authors violated the rebuttal policy by updating their project website with additional images and videos, but I'd recommend leniency given that this did not appear to materially affect the reviews.